# Determination and Quantification of the Distribution of CN-NL Nanoparticles Encapsulating Glycyrrhetic Acid on Novel Textile Surfaces with Hyperspectral Imaging

**DOI:** 10.3390/jfb11020032

**Published:** 2020-05-20

**Authors:** Kudirat A. Obisesan, Simona Neri, Elodie Bugnicourt, Inmaculada Campos, Laura Rodriguez-Turienzo

**Affiliations:** IRIS Technology Solutions S.L., Parc Mediterrani de la Technologia, Avda.Carl Friedrich Gauss No. 11, Castelldefels, 08860 Barcelona, Spain; kabidemi@iris.cat (K.A.O.); sneri@iris.cat (S.N.); ebugnicourt@iris.cat (E.B.); icampos@iris.cat (I.C.)

**Keywords:** hyperspectral imaging, principal component analysis, spectroscopy, chitosan, pullulan, partial least squares regression, nir, actives substances, cn-nl/ga

## Abstract

Chitin Lignin nanoparticles (CN-NL), standalone and encapsulating glycyrrhetic acid (GA), were applied on novel substrates for textiles to obtain antibacterial, antioxidant properties. Their homogeneous application is an important parameter that can strongly influence the final performance of the investigated textiles for its cosmetic and medical use. In this paper, hyperspectral imaging techniques combined with chemometric tools were investigated to study the distribution and quantification of CN-NL/GA on chitosan and CN-NL on pullulan substrates. To do so, samples of chitosan and pullulan impregnated with CN-NL/GA and CN-NL were analysed through Short Wave Infrared (SWIR) and Visible-Near Infrared (VisNIR) hyperspectral cameras. Two different chemometric tools for qualitative and quantitative analysis have been applied, principal component analysis (PCA) and partial least square regression (PLSR) models. Promising results were obtained in the VisNIR range, which made it possible for us to visualize the CN-NL/GA compound on chitosan and CN-NL on pullulan substrates. Additionally, the PLSR model results had determination coefficient ($R_{C}^{2})$ for calibration and cross-validation ($R_{CV}^{2})\;$ values of 0.983 and 0.857, respectively. Minimum values of root-mean-square error for calibration (RMSEC) and cross-validation (RMSECV) of CN-NL/GA were 0.333 and 0.993 g, respectively. The results demonstrate that hyperspectral imaging combined with chemometrics offers a powerful tool for studying the distribution on chitosan and pullulan substrates and to quantify the content of CN-NL/GA compounds on chitosan substrates.

## 1. Introduction

Chitin is a long-chain polymer of N-Acetylglucosamine [1]. It is a primary component of cell walls in fungi and the exoskeletons of arthropods, and it is the most common biomass waste compound after cellulose. Chitin itself presents interesting characteristics, such as biodegradability, predictable degradation rate, structural integrity and biocompatibility, which makes it an ideal candidate to produce polymeric tissues.

Chitin can assemble in the form of nanofibrils (CNs) due to a hierarchical structure, and it represents an interesting new opportunity for the pharmaceutical, cosmetic and textile sectors, thanks to their ability to complex several active ingredients by ionic charges and facilitate their delivery across the skin.

Lignin is a class of complex organic polymers rich in phenolic functional groups obtained from lignocellulosic biomass [2]. It has been reported in the literature that it can be produced as nanostructure and further complexed with CN nanofibrils. Electropositive chitin nanofibrils (CN) can be combined with electronegative nanolignin (NL), leading to microcapsule-like systems suitable for entrapping both hydrophilic and lipophilic molecules. The potential of CN-NL is that their structure can be controlled at the nanoscale, resulting in several interesting properties, such as antibacterial, anti-inflammatory, cicatrizing, and anti-aging effectiveness. Furthermore, CN-NL has also been studied as a carrier for glycyrrhetic acid (GA), a biomolecule with anti-inflammatory activity obtained from liquorice plants (Glycyrrhiza), as a model of bioactive molecule [3].

On the other hand, chitosan has been widely studied as an antimicrobial agent for preventing and treating bacterial and fungal infections. It is obtained commercially from shrimp and crab shell chitin by alkaline deacetylation 2-4 (NaOH, 40–50%). In addition, chitosan has shown cell proliferative activity essential for efficient healing. The blending of chitosan with other biopolymers along with surface modification has shown promising results in wound healing [4]. Likewise, pullulan biomaterials also enable structures for tissue regeneration, ultimately improving quality of life when used in medical and cosmetic applications.

The growing demand for anti-aging, moisturizing, skin-whitening and protective products from sun damage coupled with population ageing and the demand for more functional ingredients, drives the cosmetic market and impels us to study chitosan and pullulan-based substrates impregnated with active nanostructures as potential candidates for skin contact applications.

Over the last few decades, near-infrared spectroscopy (NIR) has arisen as an alternative method of analysis for rapidly determining the composition of a wide variety of materials belonging from many different sectors such as agriculture, food, bioactive, pharmaceuticals, petrochemicals, textiles, cosmetics, medical applications and polymers [5]. This technology can be used for the monitoring of surface composition for many different applications, such as, the analysis of fibres and their properties and the on-line monitoring of different textile coatings [6].

In the cosmetic and biomedical sectors, where the complexity of the various physical forms (e.g., creamy, milky, oily, alcoholic, aqueous, or solid) of the excipients forming emulsions is of variable consistency, the direct analysis of the sample remains challenging. Furthermore, the increasing trend of using natural bio-based components, often expensive, push the companies to look for analytical methods able to minimize the time of analysis and to control the exact amount of components added as well as to allow characterizing the natural variability of these compounds. These types of analytical methods will enable avoid extra costs as well as being able to determine the ingredients’ effective concentration. In particular, in one study reported by Blanco et al., the use of NIR spectroscopy combined with multivariate calibration demonstrated the capability of the technique to determine the components of a cosmetic mixture and its hydroxyl value [7].

In the context of the monitoring of textured surfaces composition, an emerging technique that is attracting attention is NIR hyperspectral imaging. It is a widespread, rapid, and non-destructive analytical technology, which combines conventional imaging and infrared spectroscopy to collect spatial and spectra information from a particular object, resulting in a three-way data matrix. In more detail, a traditional optical spectrum instrument commonly presents a simplex spectrum I(l), where the HSI technique offers the two-dimensional distribution of the intensity at each pixel of the image I(x, y), which is described as I(x, y, l) or I(l, x, y), which can be considered as either a detached spatial image I(x, y) at every single wavelength (l), or a spectrum I(l) at each pixel (x, y). Therefore, the main difference between HSI imaging and other imaging techniques is that for each pixel, a whole spectrum is obtained. The obtained spectrum has the function of unveiling the hidden compositional information of that particular pixel. This characteristic is very beneficial for achieving the visualization of the distribution and the composition of different chemical components in a sample. Relying on this capability, hyperspectral imaging has witnessed tremendous growth and has been widely applied in the pharmaceutical [8], food [9], and agriculture [10] sectors; however, only a few studies have been reported for the textile sector.

One recent study applied to cosmetics reports the application of multispectral imaging in the quantification of foundation applied to the skin, evaluated by using the standard deviation of the reflectance, demonstrating its potential use for both industrial and in vivo tests [11]. Due to the large quantity of information generated by these techniques, it is necessary to combine the data structure with a powerful statistical and mathematical treatment to extract the relevant information. Chemometric tools enable all spectra in a data cube to be simultaneously analysed so that the desired information can be extracted. This approach has been applied in the literature to study the heterogeneity and quantify different components of biomass derivatives [12,13].

Moreover, thanks to the ability to provide real-time material composition, NIR and HSI are increasingly being used as an emerging solution for the online separation of waste textiles depending on the nature of the used natural or synthetic fibres, among other parameters [14].

Therefore, in this work, CN-NL and CN-NL embedding GA have been applied on the surface of pullulan and chitosan using two innovative technologies, dry powder impregnation and electrospinning, that have been tested in the framework of the PolyBioSkin project.

To the best of our knowledge, this is the first study applying hsi (in both SWIR and VisNIR range) as a monitoring system for the analysis of the distribution and quantification of CN-NL as a standalone agent or encapsulating GA on pullulan and chitosan textile substrates respectively. This study confirms the potential benefit of using a non-destructive technology that can be easily adapted as inline quality control into different production lines avoiding offline assessment, as well as time and product waste.

## 2. Materials and Methods

### 2.1. Samples

Textiles of chitosan and pullulan with a porous structure were used as substrates in this study. The final set of samples were provided by different partners involved in the PolyBioSkin Project and were obtained as follows. Nanochitin, nanolignin, and glycyrrhetic acid were supplied by Mavi Sud, Aprilia (LT), (Rome, Italy). Complexes of CN-NL entrapped with GA were prepared in powder using a gelation method and a BuchiMini B-190 spray dryer technology (Flawil, Switzerland).

In the case of a pullulan-based substrate produced by electrospinning, 8% of CN-NL complexes were co-electrospun. In the case of chitosan used as a substrate, CN-NL/GA was applied through dry powder impregnation and thanks to the use of binders. The binders applied were PEG 8000 from Sigma-Aldrich, while waxy and dewaxed bleached Shellac was purchased from A.F. Suter & Co., Ltd. Complexes of CN-NL/GA were thus prepared following previous works [15,16]. The proportion of the complexes were at least 38 g nanochitin + 6.5 g lignin + 4.4 g glycyrrhetic acid with a nanoparticle size of approximately 59 nm. The complex of CN-NL impregnation on pullulan was carried out using electrospinning technology from water by BIOINICIA (Carrer de l’Algepser, 65, Nave 3, 46,980 Paterna, Valencia). Dry powder impregnation technologies of CN-NL/GA were applied on chitosan textile by FIBROLINE SA (Parc Swen Bât 4C, 1 rue des vergers 69760 LIMONEST–FRANCE) and TEXOL (Via Corradino D’Ascanio 3 65020 Alanno Scalo Pescara Italy).

In this paper, six sets of chitosan and pullulan substrates, with an approximate size of 10 × 10 cm^2^ and 5 × 5 cm^2^ for chitosan and pullulan, respectively, were evaluated. Table 1 lists each of these samples in detail, including the grams of the active substances applied. In the case of the chitosan substrate, five sets of samples were evaluated, which were composed mainly of non-woven chitosan substrates with different quantities of CN-NL/GA as active compounds and attached with different binders together with silica. Binders were applied for better cross-linkers on the chitosan, while silica was added for better fluidity of the complex. All the set samples (containing a total of 11 samples) of the chitosan substrate had the same proportion of CN-NL/GA complexes, but different quantities of the complexes were used to impregnate the chitosan textile. Finally, set 6 consists of two samples of standalone pullulan substrate and pullulan impregnated with CN-NL.

### 2.2. NIR Hyperspectral Imaging System

The hyperspectral imaging system, HYPERA^®^, developed by IRIS Technology Solutions S.L. was used for image collection, equipped with two imaging systems one in the Short Wave Infrared (SWIR) and the other in the Visible-Near Infrared (VisNIR) range, a halogen illumination system and a computer with an image acquisition platform developed by IRIS. To acquire NIR imaging, a line-scan push broom system equipped with a moving belt was used. The wavelength of the SWIR system ranged from 850 to 1600 nm, with a resolution of 4.7 nm. For the VisNIR system, the wavelength ranged from 400 to 1000 nm, with a resolution of 2.2 nm.

Both the chitosan and pullulan substrates were placed directly on the conveyor belt for image acquisition. Prior to the image acquisition, white and dark reference images were captured for image calibration. Reflectance standards are essential for image calibration, to correct pixel-to-pixel variations arising due to inconsistencies in capture and illumination of samples [17].

Each image had approximately 102,400 and 409,600 pixels for SWIR (Bolton, MA, USA) and VisNIR cameras (Bolton, MA, USA), respectively, of which about 63% were chitosan or pullulan (substrate) pixels and 37% represented background pixels. The background information, which corresponded to the pixels representing the moving belt, was eliminated to provide the best possible difference between the sample and the background.

### 2.3. Principal Component Analysis

Multivariate data analysis is a useful technique to extract relevant and meaningful information from the hyperspectral imaging data [18,19,20]. Chemometrics was an appealing tool for this study because it allowed us to reduce the dimensionality of the data while retaining essential spectral information and classify areas of interest. In this research paper, an exploratory analysis was applied to study the distribution of active substances on chitosan and pullulan surfaces.

The principal component analysis method is a qualitative method used to reduce the original set of variables into new significant variables called principal components (PC). The principal component analysis is based on bilinear decomposition, which can be mathematically described by Equation (1):(1)$$X=U\;V^{T\;}+E$$
where *X* is the unfolded matrix (matrix containing the spectra) decomposed according to Equation (1), the loadings matrix $V^{T}$ and the scores matrix *U*. The loadings matrix *V^T^* identifies the primary sources of data variance using their chemical composition (composition loadings), which may eventually be related to the CN-NL and CN-NL/GA complexes. The scores matrix, *U*, provides sample scores for these data variance patterns. The scores in each PC were re-folded to have the original spatial organization, and one image was built for each significant PC. The first principal component carries the maximum variability of the data, while the second principal component explains all the variability not included in the first component and successively. These components are defined in such a way that they are orthogonal among them. The importance of the original variables in the definition of a principal component is represented by its loadings and the projections of the objects onto the principal components are called the scores of the objects. The performance of the PCA model was evaluated mainly by variance explained, where an optimal model should provide a high variance explained, low numbers of scores, and loading to define the distribution of the active compound.

### 2.4. Partial Least Square Regression

A calibration model for quantifying the CN-NL/GA complex was performed, applying the partial least squares regression method (PLSR) [21]. PLSR is a bilinear calibration method using data reduction by compressing a large number of measured collinear variables into a few orthogonal principal components, which are known as latent variables (LVs).

The purpose of the PLS regression is to build a linear model enabling prediction of the desired characteristic (Y) from a measured spectrum (X). The model is represented by:Y = Xb + E(2)
where Y is the response matrix (nine samples of the calibration set) of chitosan score, X is the set of matrix spectral data (calibration set × wavelength), b is the matrix of regression coefficients (wavelength × 1), E is the residual information matrix that is not explained by the LV. The best number of LVs is chosen according to the lowest root mean square error of cross-validation (RMSECV) and the lowest number of latent variables.

The quality of the calibration model was evaluated using the following statistical parameters: coefficient of determination for calibration and cross-validation ($R_{C}^{2}$ and $\;R_{CV}^{2}$), root mean square error of calibration and cross-validation (RMSEC and RMSECV). The value of *R*^2^ indicates the percentage of the variance in the *Y* variable (grams of CN-NL/GA on chitosan substrate) that is accounted for by the *X* variable (spectral data). The statistical parameters were calculated as the following:(3)$$RMSE=\sqrt{\frac{1}{n}}\overset{n}{\underset{i=1}{\sum }}{\left(y_{i}-y_{m,i}\right)}^{2}$$
(4)$$R^{2}=1-\frac{\sum {\left(y_{m,i}-y_{i}\right)}^{2}}{\sum {\left(y_{m,i}-y_{mm}\right)}^{2}}$$
where *y_i_* and *y_m,i_* are the model estimated value and the reference value for sample i, respectively, and *n* is the number of samples. As mentioned by Saeys et al., a calibration model with *R*^2^ value greater than 0.91 is considered to be an excellent calibration, while an *R*^2^ value between 0.82 and 0.90 results in good prediction. A small difference between RMSEC and RMSECV value was also important to avoid “overfitting” in the calibration model.

### 2.5. Software Tools

For statistical analysis, data pre-processing of the images was performed using routines written in Matlab R2018a. Multivariate data analysis was performed with PLS Toolbox 8.6.2 running on Matlab (version 9.4, R2018a) (The MathWorks, Inc., Natick, MA, USA).

## 3. Results

### 3.1. Spectroscopic Analysis

Chitosan substrates impregnated with CN-NL/GA

A first qualitative approach to understanding the spectral homogeneity of chitosan textiles with and without impregnated CN-NL/GA nanoparticles was evaluated on pure chitosan and chitosan impregnated with 6 g of CN-NL/GA compound (sample 2 from set 2, as described in Table 1). Figure 1 shows the normalised spectra of the CN-NL/GA impregnated chitosan samples extracted from the hyperspectral imaging, which were acquired through SWIR (Figure 1a) and VisNIR (Figure 1b) cameras, respectively.

With respect to the absorption of molecules in the NIR spectroscopic region resulting from the absorption of overtones and combination bands, types of vibrations are related to the fundamental vibrations seen in this region. Most of the overtone and combination bands involving higher frequency fundamental vibrations are CH, NH, and OH stretching. Taking this fact into account, the signal contribution due to the presence of silica during the impregnation of CN-NL/GA chitosan substrates could be considered minor. However, certain metal complexes have low-lying electronic transitions in the NIR, although this article will focus on vibrational transitions. As can be seen from Figure 1a, the spectra of the reference, and the sample with active compounds show a similar profile with a common peak approximately at 1500 nm. This signal can be related to the similar chemical structures of the substrate (chitosan) and the active compound (chitin-based complexes), which present similar fingerprints in this region. Nevertheless, differences can be detected in two waveband regions, which range from 1000–1300 nm and 1400–1600 nm (SWIR). Comparing these spectroscopy regions, differences from the intensity of the signal could be observed from chitosan reference and chitosan with CN-NL/GA. The chitosan reference spectrum shows an intensity signal higher than chitosan+CN-NL/GA spectra in the range of 1000–1300 nm. Thus, this variation of the intensity might be associated with the presence of CN-NL/GA on the chitosan substrate. On the other hand, two spectra of CN-NL/GA corresponding to different zones of the same samples show closer intensity; this was expected, since they both have the complex. Moreover, a slight deformation of the peak at 1488 nm could also be highlighted, confirming the behaviour reported in the literature, probably related to the first overtone of OH stretching [22,23]. Variations of spectra intensities in some specific areas indicate that this camera could be applied to study the distribution of this compound. Furthermore, evaluating the spectra of the chitosan and chitosan+CN-NL/GA complex provided by the VisNIR ranges (Figure 1b), similar trends could be observed in SWIR range. The chitosan reference spectrum shows an intensity signal higher than the chitosan+CN-NL/GA spectra in the range of 400–800 nm, which corresponds to the visible region that is widely used in spectroscopic techniques for pigment analysis [24]. On the other hand, two spectra of CN-NL/GA corresponding to different zones of the same samples show closer intensity; this was also expected, since they both have the complex.

To visualize and evaluate the homogeneity of CN-NL/GA on chitosan surfaces, a PCA exploratory study was carried out (Section 3.2.1).

Pullulan substrates impregnated with CN-NL

The qualitative analysis of the presence of CN-NL was also performed on the pullulan and pullulan-impregnated samples (set 6, pullulan with and without CN-NL, samples 12 and 13 respectively, as described in Table 1). The spectra channels obtained from hyperspectral imaging of SWIR and VisNIR ranges are displayed in Figure 2a,b, respectively. These spectra signals corresponded to pullulan alone and two spectra of pullulan impregnated with CN-NL complex. The two impregnated spectra were identified as zone 1 and zone 2. The SWIR spectrum indicates a clear difference between pullulan substrate used as a reference and CN-NL impregnated one. Furthermore, new bands can be observed in the spectra from pullulan+CN-NL that are directly related to the impregnated pullulan, one in the range of 1000–1300 nm and the second one at 1500 nm. Since the C-H band is among the most prominent features influencing the near-infrared spectroscopy in CN-NL, the wavelengths observed at 1078 and 1197 nm could be related to it. Additionally, variation in the signal intensity could also be observed between the two impregnated zones (zone 1 and 2 from sample 12). This variation was expected, since the quantity of CN-NL might not be able to cover the surface of pullulan (sample 12) completely. Figure 2b shows the spectra resulting from the VisNIR camera. Moreover, in this case, three spectra were compared, corresponding to pullulan reference (sample 13) and two spectra of pullulan impregnated with CN-NL (zone 1 and 2, from sample 12). Spectra channels obtained from this spectroscopy range present noisy due to light scattering characteristics in VisNIR spectroscopy range. Savitzky–Golay filters have been previously applied on this signal together with normalization as pretreatments to reduce noise and extract relevant information beyond the signal. The reference pullulan substrate shows differences with the pullulan+CN-NL in the ranges of 650–750 nm and 950–1050 nm. However, the two zones of the pullulan+CN-NL show a similar trend, although slight differences could be observed between wavebands of 650 and 750 nm.

### 3.2. Qualitative Distribution Analysis (PCA Method)

#### 3.2.1. Evaluation of Chitosan Substrates

Score Results

Hyperspectral imaging combined with PCA was conducted to study the distribution of CN-NL/GA as an active substance on chitosan substrates. In this study, the background and the insignificant pixels were eliminated, and pixels corresponding to chitosan samples were obtained, which was followed by PCA. Different preprocesses were applied to the spectral data prior to analysis, suppressing scattering effects and undesirable spectral variability. These included the application of SNV (Standard Normal Variate), detrend, Savitzky–Golay, and first and second derivatives. Optimised pretreatment was achieved after applying noise correction with a smoothing filter (Window 7, polynomial 0 and orders 2), followed by normalization of each channel and a mean centre. Images are colour-coded corresponding to infrared intensity scale bars. The homogeneity of CN-NL/GA substances was evaluated based on the contribution of each pixel to the PC1, where the deep blue colour refers to the conveyor, light blue to the chitosan substrate, and the red colour shows the highest intensity of the active substances.

Figure 3a shows the RGB images of samples evaluated with the hyperspectral imaging, corresponding to two different sets of samples. These samples were chitosan impregnated with 2.68 and 6 g of CN-NL/GA compound for set 1 (sample 1) and set 2 (sample 2), respectively. Samples were evaluated with both SWIR and VisNIR ranges, and the score results of the PCA models are illustrated in Figure 3b,c.

In the case of sample 1 (set 1), the first principal component scores were chosen to discriminate between the chitosan substrate and CN-NL/GA compound, since it had the major contribution of the original information. This component described approximately 78.03% of the original information for the SWIR and 72.13% for the VisNIR cameras. In the overall images, both SWIR and VisNIR (Figure 3b,c) cameras showed a clear difference between the substrate and CN-NL/GA compound. The chitosan substrate in the absence of CN-NL/GA can be identified by the blue colour with negative values. The regions with CN-NL/GA compounds are yellow or red, indicating a change in the intensity of the reference peak. A more intense colour can be observed from the VisNIR camera, confirming the results obtained previously. Indeed, CN-NL/GA absorbs in the VisNIR range due to their pigmentation. Based on these results, hyperspectral cameras at SWIR and VisNIR spectroscopy ranges can be valuable for visualization and to study the presence of CN-NL/GA + silica distribution on chitosan substrates.

Furthermore, the score results obtained from the second sample (Set 2) from both the SWIR and VisNIR cameras are reported, and they confirmed the previous results. Indeed, in both cases, the first components explained the maximum variation of the original information with scores of PC1 78.47% and 72.34% for the SWIR and VisNIR cameras, respectively. With both cameras, we could detect a clear difference between the chitosan and chitosan impregnated with CN-NL/GA compounds. Again, higher signal intensity can be viewed from the VisNIR camera, which is related to the CN-NL/GA pigment being more intense in the visible range. The elaboration of the data related to the chitosan substrate used as a reference, results in blue colour, reinforcing the accuracy of the previous results. Based on the overall PCA score images obtained for both hyperspectral imaging cameras, this technique could probably be useful for studying the homogeneity distribution of active compounds on chitosan substrates.

Another set (set 3) of samples with chitosan and chitosan impregnated with CN-NL/GA compounds attached to waxy bleached Shellac was also studied. Three quantities of the CN-NL/GA compound—5, 5.4 and 7 g for samples 3, 4, and 5, respectively—were evaluated (see Figure 4a). In this set of samples, SWIR was also studied to see if the presence of the waxy bleached Shellac as a binder enhanced the NIR spectroscopy signal, which may improve the correct visualization of the distribution of CN-LG/GA compounds. The pretreatment applied was a smoothing filter (Window 7, polynomial 0 and orders 2) together with normalised and mean centre, which allows the elimination of the noise on the spectra and the correction of the scattering effect that is typical in NIR radiation. Hypercube data was evaluated with PCA, and the scores for PC1, explaining the maximum and relevant variability from the original information, are displayed in Figure 4b,c. The explained variance for each of the PC1s for SWIR and VisNIR is 74.15% and 61.97%, respectively. However, the results attained from the score images suggested that SWIR might not be suitable for studying the distribution on the chitosan substrate. The SWIR score results (see Figure 4b) show an incoherent output assigning different colours to CN-NL/GA-impregnated sample 5 compared to sample 3 and 4 (both with CN-NL/GA). These variations cannot be associated with the quantities of the CN-NL/GA compound, bringing us to the conclusion that SWIR might not be suitable to study the homogeneity of CN-NL/GA compound due to its lack of sensitivity for these specific compounds. More promising results were achieved from the VisNIR camera (Figure 4c). The blue colour indicates the reference, the two samples containing similar amounts of CN-NL/GA (Samples 3 and 4; 5 and 5.4 g of CN-NL/GA, respectively) can be identified by the red colour spotted with green. This colour variation can be ascribed to the impregnation technique used, since these quantities of active compounds might not be able to cover the surface completely. Additionally, when the quantities of CN-NL/GA were higher (Sample 5; 7 g of CN-NL/GA), the red colour, which is directly related to the presence of CN-NL/GA, is clearly more homogeneous, indicating that the impregnation technique is more accurate due to the higher amount of actives. These results demonstrate that HSI in VisNIR range can be applied not only for the detection of the presence of the active compounds, but could also be considered as a powerful tool for the inline quality control of the impregnation step. Chitosan with a high quantity of CN-NL/GA could be observed with more intensity (see sample 5 in Figure 4c). Furthermore, the analysis showed that samples 3 and 4, with 5 and 5.4 g of the active substance, respectively, exhibit similar distributions, indicating the heterogeneity of the distribution of CN-NL/GA compounds. This result demonstrates that hyperspectral imaging in the range of VisNIR could be a suitable technique for the detection of the composition and distribution of actives (CN-NL/GA) on chitosan substrate.

Loading Results

An important aspect of NIR hyperspectral image analysis is spectral interpretation. A preliminary PCA analysis, after applying smoothing together with normalised and mean centre, showed that the first component gives interesting details related to the chemical composition of the substrate, as well as the distribution of CN-NL/GA on chitosan. Afterward, a PCA model was developed with samples, where we have the chitosan substrate alone and chitosan impregnated with 6 g of CN-NL/GA compounds. The loading results from the PCA analysis do not give information on the standalone compounds of chitosan and CN-NL/GA, but provide information about mixed signals. Taking into account that better visualization of the distribution of CN-LN/GA compounds was obtained from the VisNIR spectroscopic range, the loading results were based on those regions.

The first loadings that explain the maximum amount of information for this region are approximately 73% of the original information, as depicted in Figure 5.

A peak in the range of 400–700 nm gives a positive contribution for the distribution of chitosan and chitosan encapsulated with CN-NL/GA. This was expected, since these regions are associated with the visible region, a typical range for many pigments [25]. Furthermore, considering the approximate proportion of the CN-NL/GA complex (68 g nanochitin + 3.5 g nanolignin + 4.5 g of glycyrrhetic acid, as mentioned in Section 2.1), chitin has the majority composition and signal contribution in the VisNIR spectroscopy. The waveband at 400–700 nm due to pigment of CN-NL/GA complex might be due to chitin composition. As mentioned in [26], chitin obtained from crustacean shells could contain pigments of fungi; thanks to the presence of these pigments, the visualization of CN-NL/GA complex could be possible when applying chemometric tools. Additionally, peaks at wavebands near 850 and 950 nm could be associated with the vibration of the NH and OH (third overtone), respectively, of the chitin substances.

#### 3.2.2. Evaluation of Pullulan Substrates

Score Results

The distribution of CN-NL compounds was also evaluated on pullulan textiles used for cosmetic applications. NIR spectroscopy has a high level of sensitivity to noise, as observed in Figure 2. However, the PCA model was applied to this data, since chemometric tools are characteristic in reducing and extracting the relevant information in addition to the signals. Taking into account all these factors, the PCA model was applied on the pullulan substrate and the impregnated substrates for both SWIR and VisNIR spectroscopy ranges. Different data pretreatments were tested to eliminate some sources of variation related to light scattering, or at least to reduce the noise. The data pretreatments applied included smoothing, together with normalised and mean centre to centralise all the data. For a better interpretation of the results, images obtained from SWIR and VisNIR ranges were adjusted according to the same intensity. Figure 6 illustrates pullulan samples analysed through hyperspectral imaging, together with the scores for the PC1 images obtained from SWIR and VisNIR cameras. These samples included pullulan alone (used as a reference, sample 13) and pullulan impregnated with CN-NL complex (sample 12) (see Figure 6a). The explained variance of the PCA models built for each of the ranges were 75.82% and 74.4%, respectively, demonstrating that the scores of the first components described the maximum amount of information of the original hypercubes.

The discrimination of CN-NL substances on the pullulan substrate was also evaluated based on the PCA model. The score results from the SWIR range can be observed in Figure 6b, where the deep blue colour is assigned to the conveyor belt, while the reference pullulan substrate sample (see sample 13) can be identified by its light blue colour with negative score values. Nevertheless, the pullulan substrate impregnated with active substances is yellow, derived from changes in the intensity in that region. As illustrated in Figure 6b, SWIR was not able to discriminate between the presence of CN-NL on the pullulan substrate. Furthermore, the PC1 score image results of VisNIR are also displayed in Figure 6c, and differences can be observed in the pullulan with and without the active substances. In this case, a clear change can be observed from the reference and pullulan impregnated with CN-NL substances. The reference substrate can be identified by its red colour, and the pullulan impregnated with CN-NL compounds can be identified by their yellow or light blue colours. Thus, this work proved the possibility of applying hyperspectral imaging techniques in the range of VisNIR for the detection of the composition and distribution of actives (CN-NL) on a pullulan substrate, even though the chemical composition shows a proper difference between the pullulan and the impregnated sample.

Loadings Result

To study the chemical changes that occur on pullulan substrate with respect to pullulan impregnated with CN-NL compounds, the loadings of the first principal components were evaluated. Figure 7 shows the loading plots of the PC1 for the VisNIR camera that contribute to the changes that occur between the reference substrate (pullulan) and pullulan impregnated with the active compound, respectively. The loading variables displayed in Figure 7 show two broadening bands that contribute to these changes found both in positive and negative regions. Loading results from the display variables ranged between 550–650 nm and 950 nm. The wavebands in the range of 550–650 nm were mainly associated with the visible range. Additionally, the peak at the waveband near 850 nm might be related to the OH stretching of the third overtone from the chitin substances.

### 3.3. Quantitative Distribution Analysis (PCA Method)

Taking into account the promising results obtained from the PCA model to study the distribution of the CN-NL/GA compounds on chitosan substrate with VisNIR, PLS regression models were implemented to quantify the CN-NL/GA compound on chitosan substrate with hypercubes obtained with this camera. To do so, nine samples, including chitosan alone and chitosan+CN-LN/GA, listed previously in Table 1, were used. The Chitosan+CN-NL/GA complex used included samples from set 1 (sample 1), set 2 (sample 2), set 3 (sample 3), set 4 (samples 5, 6, and 7) and set 5 (samples 8, 9 and 10), with a range of 0–8 g for the CN-NL/GA compound. The mean spectra for each of the samples are plotted in Figure 8a, where slight differences can be observed from one of the samples, which can be associated with chitosan substrate without CN-NL/GA compound. Slight variation in the intensity of CN-NL/GA compound signals can also be perceived between the range of 400–800 nm. Although these changes in absorbance can seem irrelevant, the differences among them are suitable enough for a multidimensional analysis such as PLS. With the aim of building robust models capable of predicting totally unknown samples, the original dataset of spectra was used to perform the training set as well as an internal cross-validation of the model.

To build the PLS regression model, the mean NIR spectra from each of the samples were correlated with the grams of CN-NL/GA impregnated on the chitosan substrate. All spectroscopic ranges were selected to build the calibration. Table 2 shows several pretreatments tested together with coefficients of determination calibration, leave-one-out-cross-validation (R^2^_Cal_ and R^2^_cv_), and the root mean square error for calibration and cross-validation (RMSEC, RMSECV) respectively for each of the pretreatments tested. In all cases, three latent variables were needed to build the regression models. Based on the results in Table 2, when no pretreatment was performed, the values of RMSEC and RMSECV were 1.335 and 2.435, respectively, indicating that large prediction errors would be caused if data was directly applied to the spectra measured. For smoothing the spectra, Savitzky–Golay (SG) was applied to the calibration model together with normalisation, SNV, baseline, and mean centre to improve the model. Among all the pretreatments, preprocessing applying smoothing together with baseline and mean centre attained the minimum values of RMSEC and RMSECV, which are 0.333 and 0.993 g of CN-NL/GA, respectively. Additionally, high values for the coefficient of determination both for calibration and cross-validation were also assigned for this pretreatment, with the values obtained being 0.983 and 0.857, respectively.

## 4. Conclusions

This study shows the possibility of applying hyperspectral imaging techniques in both the SWIR and VisNIR ranges, coupled with chemometric tools, to study the distribution of CN-NL/GA and CN-NL on chitosan and pullulan substrates, respectively, and the quantification of CN-LN/GA on chitosan substrate. The overall results obtained from the PC score images on the chitosan textiles indicate that the VisNIR spectroscopy range is a useful support for the correct visualization of the distribution between chitosan with CN-NL/GA complex. Thus, the quantification of CN-NL/GA on chitosan using the PLS model developed for this purpose, shows a satisfactory result with an accuracy of 98.3%, indicating that the model is suited to the quantification of CN-NL/GA compounds on chitosan. In addition, the pullulan substrate also showed promising results for visualizing the presence of the CN-NL complex; however, further studies should be performed on this type of textile. The results show the feasibility of developing a simple, rapid, non-destructive, and reliable method for determining the homogeneity distribution of CN-NL/GA nanoparticles on chitosan and CN-NL on pullulan substrates after multivariate analysis. This approach can be easily applied for the in-line monitoring of the quality of the textiles used for cosmetic and biomedical applications, as well as for quality control and optimization of the impregnation process, in order to reduce time and resource waste.

## 5. Future Perspectives

Future work could involve the application of analytical reference techniques to confirm the differences in the quantity of CN-NL/GA and CN-NL complexes on different surfaces of chitosan and pullulan substrates. This chemical information would make it possible to quantify the differences in the application results observed from the HSI technique. This information could be useful to demonstrate that HSI could be a reliable homogeneity monitoring tool for the textile industry.

Furthermore, concerning the pullulan substrate, more samples of pullulan impregnated with CN-NL complex should be evaluated with the HSI techniques. These could include samples with different quantities of CN-NL impregnated on pullulan surfaces to ensure that the VisNIR range is suitable to study the presence of chitin complex and its distribution on the pullulan surface.

## Figures and Tables

**Figure 1 jfb-11-00032-f001:**
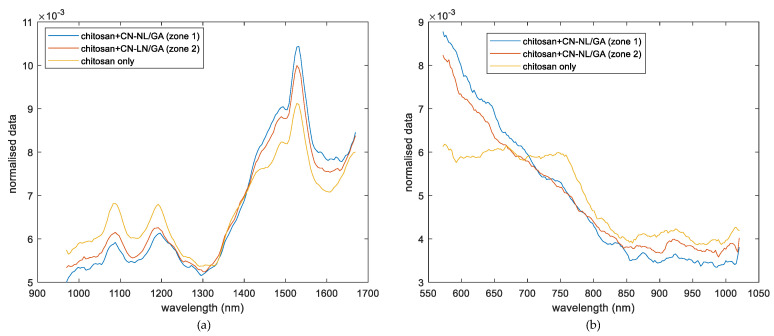
Comparison of hyperspectral imaging signal obtained from (**a**) SWIR range, a spectrum corresponding to pure chitosan substrate and two spectra of chitosan impregnated with CN-NL/GA in two different zones from the same sample (sample 2) and (**b**) VisNIR range, a spectrum corresponding to pure chitosan substrate and two spectra of chitosan impregnated with CN-NL/GA in two different zones from the same sample (sample 2).

**Figure 2 jfb-11-00032-f002:**
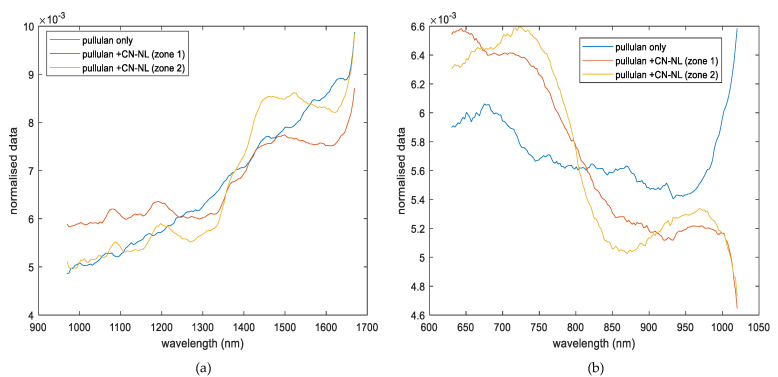
Comparison of hyperspectral imaging signal obtained from (**a**) SWIR range, corresponding to the pullulan substrate (sample 13) and spectra from pullulan impregnated with CN-NL in two different zones from sample 12 (set 6), and (**b**) VisNIR range, corresponding to the pullulan substrate (sample 13) and spectra of pullulan impregnated with CN-NL in two different zones from sample 12 (set 6).

**Figure 3 jfb-11-00032-f003:**
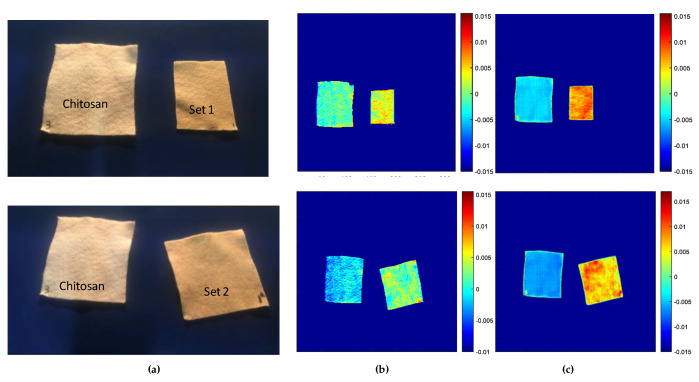
PCA score images for the first principal components: (**a**) Picture of the chitosan and chitosan impregnated with CN-NL/GA (set 1, top images: chitosan with 2.68 g CN-NL/GA and set 2, down images: chitosan with 6 g of CN-NL/GA); (**b**) PC1 scores image for SWIR camera; and (**c**) PC1 scores image for VisNIR camera.

**Figure 4 jfb-11-00032-f004:**
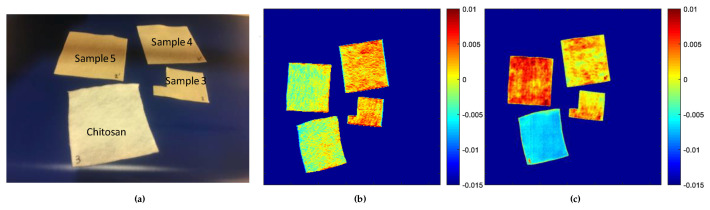
PCA scores images for the first principal components: (**a**) Picture of chitosan and chitosan impregnated with different quantities of CN-NL/GA compound (chitosan: 0 g, samples 3: 5 g; sample 4: 5.4 g; sample 5: 7 g; from set 3); (**b**) PC1 scores for SWIR camera; and (**c**) PC1 scores for VisNIR camera.

**Figure 5 jfb-11-00032-f005:**
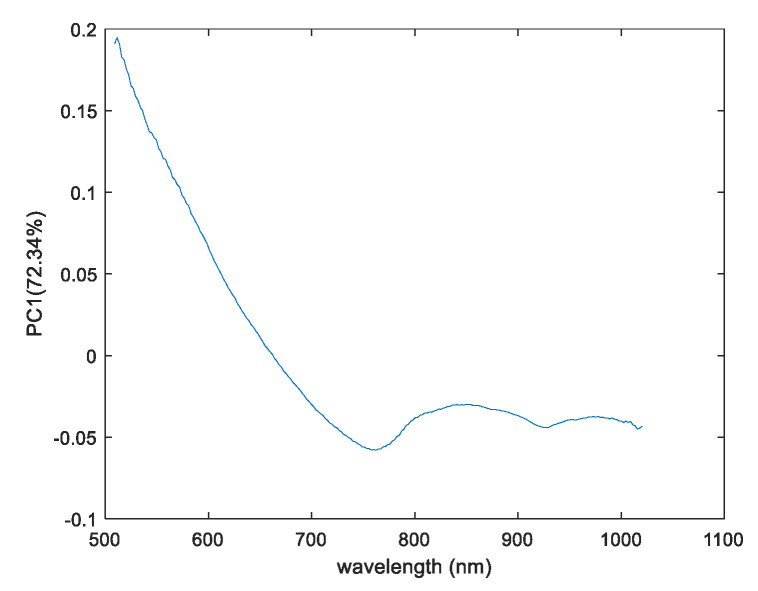
The PC1 Loadings plot result for chitosan and chitosan impregnated with CN-LN/GA for VisNIR camera.

**Figure 6 jfb-11-00032-f006:**
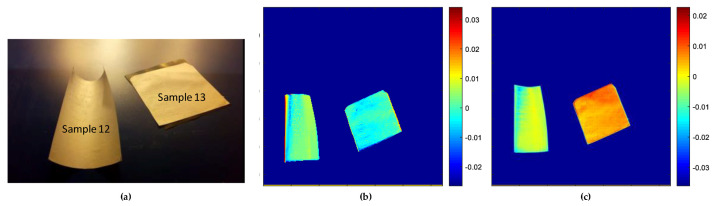
PCA scores images for the first principal components: (**a**) Picture of pullulan reference (sample 13) and pullulan impregnated with CN-NL complex (sample 12); (**b**) PC1 scores for SWIR camera; and (**c**) PC1 scores for VisNIR camera.

**Figure 7 jfb-11-00032-f007:**
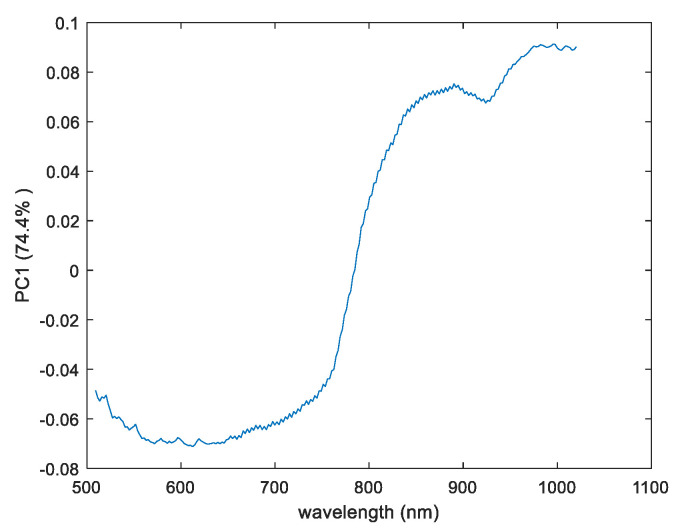
The PC1 for loadings plot result of pullulan and pullulan impregnated with CN-NL for the VisNIR camera.

**Figure 8 jfb-11-00032-f008:**
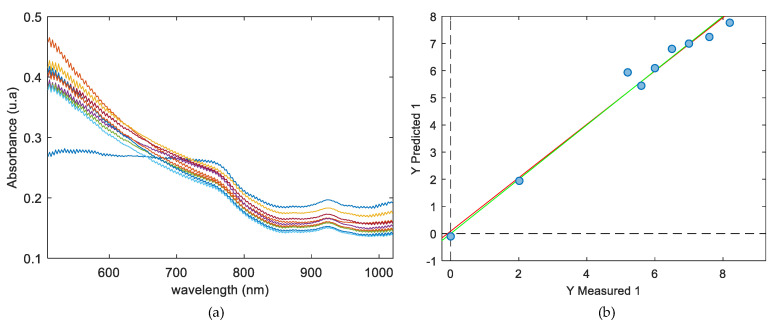
(**a**) Mean spectral for each of the samples; (**b**) PLS regression results for CN-NL/GA quantification, showing the actual grams of CN-NL/GA vs. the predicted grams of CN-NL/GA complex.

**Table 1 jfb-11-00032-t001:** List of chitosan and pullulan substrates selected for hyperspectral imaging evaluation. Sets 1 to 5 are chitosan substrates with different quantities of CN-NL/GA complex, and set 6 corresponds to pullulan substrate and pullulan with chitin nanofibrils.

Substrates	Set of Samples	Number of Samples	Substrate Base + Active Compounds + Binders
Chitosan	Set 1	Sample 1	Chitosan + 2.68 g CN-NL/GA + 2% Silica
	Set 2	Sample 2	Chitosan + 6 g CN-NL/GA + 2% Silica
	Set 3	Sample 3 Samples 4 Sample 5	Chitosan + 5 g (50% CN-NL/GA + 50% waxy bleached Shellac) + 2% Silica Chitosan + 5.4 g (50% CN-NL/GA + 50% waxy bleached Shellac) + 2% Silica Chitosan + 7 g (50% CN-NL/GA + 50% waxy bleached Shellac) + 2% Silica
	Set 4	Sample 6 Sample 7 Sample 8	Chitosan + 2.016 g (50% CN-NL/GA + 50% dewaxed bleached Shellac) + 2% silica Chitosan + 5.2 g (50% CN-NL/GA +50% dewaxed bleached Shellac) + 2% silica Chitosan + 5.6 g (50% CN-NL/GA + 50% dewaxed bleached Shellac) + 2% silica
	Set 5	Sample 9 Sample 10 Sample 11	Chitosan + 6.5 g (50% CN-NL/GA + 50% PEG) + 2% silica Chitosan + 7.6 g (50% CN-NL/GA + 50% PEG) + 2% silica Chitosan + 8.2 g (50% CN-NL/GA + 50% PEG) + 2% silica
Pullulan	set 6	Sample 12 Sample 13	Pullulan + 8% CN-NL Pullulan only

**Table 2 jfb-11-00032-t002:** Regression statistics results of latent variables, LVs; coefficient of determination for calibration and cross-validation (R^2^_Cal_ and R^2^_cv_); the root mean square error for calibration and cross-validation (RMSEC and RMSECV), to quantify CN-NL/GA compound on chitosan substrate.

Pretreatments	LVs	R^2^_Cal_	R^2^_CV_	RMSEC	RMSECV
non pretreatment	3	0.721	0.278	1.335	2.435
smoothing + normalize + mean centre	3	0.973	0.743	0.418	1.485
smoothing + SNV + mean centre	3	0.977	0.624	0.384	4.820
smoothing + baseline + mean centre	3	0.983	0.857	0.333	0.993

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
