# Peer review of "Determination and Quantification of the Distribution of CN-NL Nanoparticles Encapsulating Glycyrrhetic Acid on Novel Textile Surfaces with Hyperspectral Imaging"

_jfb, 2020, doi:10.3390/jfb11020032_

Round 1

Reviewer 1 Report

See comments in attached document

Reviewer 2 Report

Dear Authors,

The paper presented to review reports the study with the capability of applying Hyperspectral imaging techniques in both the SWIR and ViSNIR ranges coupling with chemometric tools to study the distribution and quantify the content of actives substances (CLA compounds) on chitosan and pullulan textile substrates.

Here are some ways on how the paper can be improved:

Introduction:

The Introduction section need a major revision with a better explanation of the study propose and proper reference included showing a small state of the art.

Results:

Results are well presented,although the graphs need a higher quality image.

Discussion:

I have no comments.

Conclusions:

I suggest to introduce a future work paragraph.

Round 2

Reviewer 1 Report

The quality of the paper was greatly improved. I just have a few last suggestions:

  • Currently, the references seem to be positioned at the end of the paragraphs. The standard I know is to locate the reference at the end of the first sentence of a series of successive sentences referring to the same source(s). You include again the reference at the end of the first sentence if you start another paragraph with the same source.
  • There are some typos and grammatical errors left, so you should have the manuscript proof-edited.
